# Recent Advances in Aptasensing Strategies for Monitoring Phycotoxins: Promising for Food Safety

**DOI:** 10.3390/bios13010056

**Published:** 2022-12-29

**Authors:** Hamed Zahraee, Atiyeh Mehrzad, Khalil Abnous, Chih-Hsin Chen, Zahra Khoshbin, Asma Verdian

**Affiliations:** 1Targeted Drug Delivery Research Center, Pharmaceutical Technology Institute, Mashhad University of Medical Sciences, Mashhad 9177948564, Iran; 2Pharmaceutical Research Center, Pharmaceutical Technology Institute, Mashhad University of Medical Sciences, Mashhad 9177948564, Iran; 3Department of Medicinal Chemistry, School of Pharmacy, Mashhad University of Medical Sciences, Mashhad 9177948564, Iran; 4Department of Food Safety and Quality Control, Research Institute of Food Science and Technology (RIFST), Mashhad 1314983651, Iran; 5Department of Food Biotechnology, Research Institute of Food Science and Technology (RIFST), Mashhad 1314983651, Iran; 6Department of Chemistry, Tamkang University, New Taipei City 25137, Taiwan

**Keywords:** phycotoxin, aptamer, biosensor, nanomaterial, on-site detection, portable

## Abstract

Phycotoxins or marine toxins cause massive harm to humans, livestock, and pets. Current strategies based on ordinary methods are long time-wise and require expert operators, and are not reliable for on-site and real-time use. Therefore, it is urgent to exploit new detection methods for marine toxins with high sensitivity and specificity, low detection limits, convenience, and high efficiency. Conversely, biosensors can distinguish poisons with less response time and higher selectivity than the common strategies. Aptamer-based biosensors (aptasensors) are potent for environmental monitoring, especially for on-site and real-time determination of marine toxins and freshwater microorganisms, and with a degree of superiority over other biosensors, making them worth considering. This article reviews the designed aptasensors based on the different strategies for detecting the various phycotoxins.

## 1. Introduction

Phycotoxins, likewise named shellfish toxins or algal biotoxins, are toxic and active metabolites. They are a member of extremely varied classes of small- to medium-sized organic compounds (300–3500 Da), mostly generated by dinoflagellates, diatoms, and cyanobacteria [1]. These toxins can behave like vectors that can move and accumulate in the water, fish, and shellfish as seafood is considered a severe threat to consumers due to some neurotoxic and diarrhetic effects or amnesic harm [2,3]. More than 1000 phycotoxins have been found and classified into eight groups based on their structure: azaspiracids (AZAs), brevetoxins (BTXs), cyclic imines, domoic acid (DA), okadaic corrosive (OA), pectenotoxins (PTXs), sax-itoxins (STXs), and yessotoxins (YTXs). According to the chemical classes, they are categorized into amino acids (e.g., DA), tetrahydropurines (e.g., STXs), and polyketides (e.g., OA, AZAs, PTXs, BTXs, and YTXs) with several analogs. Likewise, phycotoxins are further classified into hydrophilic (STXs and DA) and lipophilic (AZA, YTX, PTX, and OA) toxins based on their capacity to dissolve in water and other natural solvents [4]. The effects of intoxication or the capacity of phytotoxins to dissolve in water or other natural solvents can also be used to categorize them. Five distinct classifications of human shellfish harm have been described: paralytic shellfish poisoning (PSP) affected by STXs, neurotoxic shellfish poisoning (NSP) affected by BTXs, diarrheic shellfish poisoning (DSP) affected by the OA group of toxins and diaphysis toxins (DTXs), amnesic shellfish poisoning (ASP) affected by DA, and azaspiracid shellfish poisoning (AZP) affected by AZAs [1].

Palytoxin (PTX) and its analogs are considered the most potent marine toxins with the potential to cause rapid cardiac failure and death within minutes of intoxication [5]. A common polyether toxin is OA, and one of the alkaloid compounds is the well-known tetrodotoxin (TTX) [6]. TTX preferentially blocks Na+ channels on nerve cell membranes, which results in nerve paralysis and respiratory failure and can result in the poisoning and death of an adult [5]. Identically, OA toxin can cause carcinogenic or teratogenic effects [7]. There are five different types of cyanotoxins, which can cause gastrointestinal, neural, hepatic, or dermal toxicity, including (a) hepatotoxins (cylindrospermopsin (CYN), microcystins (MCs), and nodularin (NOD)), (b) neurotoxins (anatoxin-a (ATX), STX, and β-N-methylamino-L-alanine (BMAA)), (c) dermatotoxins, and (d) cytotoxins [8].

Concerns regarding phycotoxin poisoning have grown recently, grabbing the attention of consumers and food safety organizations. Some reports of the rising the level of shellfish toxins in July 2020 in Ireland, and the collecting of shellfish by the Zealandia governments, are due to elevated levels of Paralytic Shellfish Poisoning (PSP) toxins found in shellfish samples. Similarly, the Ocean and Fishery Bureau issued a warning notice on the eating of Haihong in Hebei, China, in May 2021 based on test findings showing that the paralytic shellfish toxin contained in Haihong’s body had greatly surpassed the safety limit [3]. Considering the global impact of these toxins on humans, livestock, and pets, and with the lack of effective cure for the diseases caused in mind, the only possible viable way is to prevent their entry into the food chain [6,9]. As such, it seems to be necessary to improve detection methods. Therefore, achieving specific and sensitive methods is the most effective way to prevent the emergent issues caused by this kind of biotoxin [8]. Conventional detection methods such as thin-layer chromatography and high-performance liquid chromatography are easy to be operated. Still, there are significant drawbacks associated with them, including cost because of professional instruments, unsuitability for on-site monitoring, the need for highly trained personnel, and time requirements owing to complicated pretreatment [10,11]. In addition, well-known immunological alternative approaches for detecting marine toxins are available, such as neurotoxicity, immunological interactions, hepatotoxicity, cytotoxicity, and enzymatic activity. Meanwhile, the first and most common method, cytotoxicity-based mouse bioassay, has ethical drawbacks in addition to its poor specificity, high cost, low sensitivity, poor accuracy, prolonged processes, and high variability. These difficulties limit the widespread application of immune assay platforms [12].

Given that the previous detection methods exhibit low specificity and ethical or technical limited applicability in detecting these compounds, inventing new identification strategies for monitoring phytotoxins seems necessary to ensure food and health safety. Then, extensive research was conducted on the use of alternative methods with the potential for reliable, timely, easy-to-use, and low-cost identification, as well as sensitivity, specificity, and reproducibility equal to or greater than conventional analysis methods. As a result, biosensors were introduced as a powerful alternative to conventional analysis methods [13]. Biosensors, particularly antibody-based immunosensors, and aptamer-based aptasensors, have attracted widespread interest and played critical roles in the field of fast detection in recent years, owing to their advantages of rapid response, high sensitivity and selectivity, and easy operation [3]. However, several attempts have been undertaken up to this point to facilitate flexible and effective and sensitive sensors as alternatives to conventional techniques employing nanomaterials with unique properties. For instance, single-walled carbon nanotubes (CNTs), titanium dioxide nanotubes, graphene oxide (GO) nanosheets, molybdenum disulfide (MoS_2_) nanosheets, graphene hydrogels, gold nanoparticles (AuNPs), lanthanide nanoparticles, and quantum dots (QDs) can be practical signal amplifiers for improving the sensitivity and selectivity of sensors [12,14,15,16].

Aptamers as the single strands of DNA or RNA oligonucleotides are employed with the role of bioreceptor in biosensing assays that can bind to specific targets with high affinity [13]. Hence, aptamers were recently invented to detect hazardous small compounds such as antibiotics, pesticides, insecticides, pollutants, poisons, and allergies. Aptamer technology is a promising technique for detecting these compounds from a variety of matrices [17]. Aptamers were considered to be viable even when combined with the many nanomaterials that may be utilized to produce practical, extensive, lightweight, and adaptable devices and sensors [18,19]. Besides, aptamers diverge significantly from antibodies in several important ways, including simple synthesis, batch-to-batch consistency, simplicity in labeling, low cost, and excellent stability [20]. Hence, the widespread selection of aptamers against many analytes is specifically made to develop powerful aptamer-based biosensing platforms (aptasensors). Over the last few years, aptasensors have demonstrated great potential in rapid and sensitive target detection. They have promoted sensing strategies with controllable response time, high selectivity, low immunogenicity, stable, low-cost, and biocompatible, and considered high-throughput phycotoxins detection methods [19,21].

In this context, Dolin et al. 2021 focused on both currently available methods (ELISA) and lateral flow membrane-based immunoassays (LFIA) and, recent advances in innovative strategies for some marine biotoxins (okadaic acid, DTXs, saxitoxin, and domoic acid (DA)) [22]. In 2022, Majdinasab and Marty conducted a review of current achievements in different biomarker detection utilizing only electrochemical aptasensors [23]. In 2022, Kadam et al. conducted a review of aptameric biosensors developed to detect various harmful poisons in food, water, human fluids, and the environment [17]. As the latest assessment, in 2022 Zhao et al. reviewed the latest research progress of aptasensors toward shellfish toxins from 2016 to 2021 in terms of construction principles, signal transduction techniques, and response types [3].

The preceding reviews, which are helpful in outlining and supporting readers in understanding that various aptasensors have great potentials to be employed to detect marine biotoxins targets, are noticeably quite different from the current review. In contrast to prior research, our study reviews the different aptasensor types designed to detect potentially phycotoxins contaminating water and food matrices a compilation from 2013 to 2022). Herein, an inclusive study is given with emphasis on the new approaches for the usage of aptamers in biosensor development for the on-site and real-time identity of some important phycotoxins based on their pollutant origins.

## 2. Overview of Developed Aptasensors for Phycotoxins

To date, an increasing number of aptasensors has been introduced for the different categories of marine biotoxins, including polyether toxins (PTX and OA) [23,24] cyanotoxins (BTX-2 and MC) [8,9] alkaloid toxins (TTX, ATX-a, STX, and gonyautoxin (GTX)) [25,26,27] and also, cyclic imines (methyl spirolide G (SPX-G)) [28]. A summary of the latest studies on monitoring phycotoxins by aptasensors is given in Table 1.

## 3. Monitoring of Phycotoxin Pollutants in Environmental and Drinking Water

In simple words, biotoxins known as phycotoxins, which are generated by harmful algal blooms species (HABs), can bioaccumulate in the tissues of mussel shellfish during the time and, when taken by people, can result in severe/fatal shellfish poisoning syndromes. Phycotoxins pose a significant hazard to various marine life, including fish, seabirds, crustaceans, and marine mammals [53]. Additionally, cyanoHAB toxins in drinking water reservoirs may pose major health dangers to the public. The WHO recommends a limit of 1 g/L for the most harmful cyanotoxins, microcystin-LR, and saxitoxins, which are present in drinking and recreational waters and block protein phosphatase, causing liver failure and hepatic hemorrhage [54]. Additionally, dissolved microcystin may stay in water for a long time without any adverse effects [55].

### 3.1. Microcystin-Leucine-Arginine (MC-LR)

Microcystins (MCs) are a class of hepatotoxins generated by cyanobacteria, the most well-known of which is microcystin-LR (MC-LR). Poisoning with MC-LR and other MCs can occur through drinking tap water or having skin contact with contaminated lake water or river water [29]. MC-LR can inhibit the activities of protein phosphatases type 1 (PP1) and type 2A (PP2A) that cause hepatotoxicity. This compound, even in deficient concentrations, brings lots of clinical effects, such as skin irritation, vomiting, diarrhea, and functional and structural disturbances [56]. The clinical effects of microcystin poisoning depend on the route, the level, and the mixture of components involved in the exposure. Irritating effects result in inflammatory reactions in the skin, respiratory system, and gastrointestinal system at low to moderate exposure levels. Higher exposure levels, particularly oral exposures, result in severe liver damage. Increasing in concentrations of liver enzymes in the blood and liver swelling are the signs of liver damage that can appear within minutes to hours. These symptoms may be followed by vomiting and diarrhea, which may worsen with time. Variable central nervous system (CNS) is the final stage of lethal poisoning that may be associated with dysfunction, recumbency, and coma [57,58].

#### A Retrospect of Past Studies on Microcystins-LR (MCs) Aptasensor Platforms

Several analytical and biochemical screening methods are currently available, including bioassays, immunoassays, chromatographic techniques, and biosensors. The amount of aptasensing investigations on MC-LR is made clear by the studies that are now accessible on the subgroups of marine toxins. Different types of aptasensors have been developed to quantify MC-LR, including colorimetric, fluorescent, and electrochemical ones. However, no LC-based aptasensor has been introduced for MC-LR. Nevertheless, applying the various nano matters, enzymes, aptamer structures, and intercalating dyes provides a wide variety of portable and wearable aptasensors for the sensitive detection of MC-LR. In the following, we will review the latest biosensors developed for MCs detection. Table 2, Table 3 and Table 4 are also supplied after each section to demonstrate which aptamer (based on the lowest (kd) value) is the best and most efficient for recognizing various phycotoxins. As can be seen, each aptamer’s 5′ or 3′ ends has undergone a number of modifications in addition to being given certain necessary qualities (Table 2, Table 3 and Table 4).

In this context, Lin et al. [31] developed an electrochemical impedance aptasensor for the detection of MC-LR. Alkanethiol group aptamers were assembled on the Au electrode only through the Au-S interaction. On the strength of the covalent bonds, a monolayer of the aptamers formed on the electrode surface, which prevented the access of the external redox mediator [Fe(CN)_6_]^3−/4−^ to the electrode surface. In the presence of MC-LR, the binding of MC-LR and aptamers induced the change in the aptamer conformation that decreased the impedance response as a result of penetrating [Fe(CN)_6_]^3−/4−^ onto the electrode surface. The proposed aptasensor possesses satisfying selectivity and stability for detecting MC-LR toxin in real water samples. In this regard, an aptasensor was developed by Eissa et al. [14], a sensitive tag-free voltammetric aptasensor for quantifying MC-LR with high stability. The aptasensor consisted of graphene-modified screen-printed carbon electrodes (SPEs) for anchoring the DNA aptamers on the electrode through the noncovalent interactions. It caused a marked fall in the square wave voltammetric reduction (SWV) signal of the [Fe(CN)_6_]^4−/3−^ redox couple. Upon the binding of MC-LR to the aptamer, the SWV peak of [Fe(CN)_6_]^3−/4−^ increased considerably, indicating an enhancement in the electron transfer efficiency between the [Fe(CN)_6_]^3−/4−^ redox probe and the electrode surface. The aptasensor is stable for over two weeks, making it appropriate for practical application.

Herein, Xiang et al. [29] constructed a simple aptasensor for the selective detection of MC-LR based on the catalytic activity of horseradish peroxidase (HRP). The glass surface of microchannels was treated through an amidation reaction. They were dipped in ethanolic NaOH solution to acquire enough SiOH groups, and then they were modified with methylbenzene and APTES to give binding sites for glutaraldehyde (GA) as a cross-linker via the amino group. The MC-LR and HRP-labeled antibodies were attached to the immobilized aptamer strands. So, the chemical luminous substrate was catalyzed by HRP, which was detectable by a photodiode-based portable analyzer. The developed aptamer-antibody immunoassay can efficiently be a portable sensing platform to recognize distinct epitopes of MC-LR rapidly. Another attempt was made by Wang et al. [37] that introduced a colorimetric aptasensor by forming a Y-shaped DNA duplex on the surface of AuNPs. Two distinct linker strands (probe1 and probe2) were connected to AuNPs, resulting in AuNPs-probe1 and AuNPs-probe2 conjugates. In the absence of MC-LR, the linker stands were hybridized with the specific aptamer and formed a Y-shaped DNA duplex, also causing the formation of the dimeric structures of AuNPs. So, the solution color changed from red to blue, observable by the naked eye. The change in the conformation of the aptamers during the confrontation with MC-LR induced the pre-shaped AuNP dimers to dismantle into monomers. Accordingly, the solution color changed from blue to red. The aptasensor also showed no cross-reactivity with MC-LA or MC-YR, while remaining explicit only with MC-LR.

Among these, Li et al. [34] designed a highly sensitive, label-free, and time-saving colorimetric aptasensor for MC-LR detection for the first time. Aptamer-based colorimetry produces color variations through aptamer coupling of nanomaterials or enzymes catalyzed by specified substrates. This sensing approach is illustrated in Figure 1a. In their design, the MC-LR aptamer as the recognition element was added to AuNPs. The AuNPs’ accumulation was incited by the concentrated salt component because of the electrostatic screening impact. Due to the integration between Au and the nitrogen atoms of the aptamer bases, it could get involved with the surface of AuNPs. As the aptamer carries more negative charges, it can behave as a defender of the character from salt-induced aggregation. In the presence of MC-LR, folding of the aptamer happened from the random coil structure into a regular 3D one, leading to the unexposed bases and no more extended protection of AuNPs from salt-induced aggregation. So, the interparticle plasmon coupling caused the turning of the color of AuNPs into blue. The established colorimetric aptasensor was sensitive for the quantitative analysis of MC-LR in tap water and pond water samples.

As a result of the study conducted by Zhang et al. [33], a simple, rapid, real-time, and tag-free aptasensor based on a microcantilever array to distinguish MC-LR from its congeners has been proposed. Relying on their potential for measuring many physical, chemical, and even biochemical factors, the platforms based on microcantilevers are reliable for the simultaneous detection of multiple chemical or biological analytes [34]. The aptamer was covalently and directly fixed on the microcantilever surface via its thiol group by immersing the golden microcantilever array in the specific aptamer solution. In the presence of MC-LR, the surface stress of the microcantilever changed by forming the aptamer-target complex. A quadrant photodiode that indicated the quantity of MC-LR in the sample detected the declinations in the microcantilever array. Taghdisi et al. [35] constructed a novel sensitive fluorescent aptasensor for the detection of MC-LR by using CNTs, dapoxyl dye (fluorescent dye), and an aptamer (DAP-10) with affinity to dapoxyl dye. CNTs with unique properties of ultra-lightweight, high thermal and mechanical stability, and large surface area were utilized as aptamer immobilizers. In the absence of MC-LR, the dapoxyl attached to DAP-10 could not penetrate the surface of CNTs, resulting in vigorous fluorescence intensity. In the presence of MC-LR, the DAP-10 strand anchored on the surface of CNTs, and it could not interact with the DAP-10 dye. So, a feeble fluorescence response was detected. As an advantage, the fabricated fluorescent aptasensor required a short time (only 75 min) for the MC-LR detection (Figure 1b).

Recently, He et al. [38] developed a new aptasensor anchored on the solid-state nanopore to determine MC-LR. The specific aptamer and its complementary sequence (cDNA) were fixed separately on the surface of AuNPs. Even in the absence of MC-LR, the interaction between the aptamer and the cDNA strand stimulated the formation of a polymeric AuNP scaffold. Within adding MC-LR, the aptamer complexation with MC-LR separated the MC-LR-aptamer-AuNP ternary complex from the cDNA-AuNPs. Utilizing a solid-state nanopore, the ongoing bar of the MC-LR-aptamer-AuNP ternary complex was recognized. Beyond that, Abnous et al. [20] also described a unique MC-LR detection method based on an infinity-shaped DNA structure and terminal deoxynucleotidyl transferase (TdT). Figure 1c displays the sensing function of the developed aptasensor. TdT DNA polymerase can elongate ssDNAs or dsDNAs at the 3′ ends. The enzyme formed the infinity-shaped DNA structure on the electrode surface by adding a poly-T tail to the 3′-end of the particular aptamer. The infinity-shaped aptamer was split from its complementary strand in the presence of MC-LR. Subsequently, no infinity-shaped structure could construct after the introduction of TdT, resulting in more entry of the redox marker [Fe(CN)_6_]^4−/3−^ to the electrode surface. Hence, this phenomenon led to a strong signal. The proposed electrochemical aptasensor provided the highly sensitive and selective monitoring of MC-LR with no cross-reaction behavior to similar toxins, such as MC-LA, atrazine, acetamiprid, zearalenone, and AFM1.

Based on the study by Lee et al. [36], a QD-based aptasensor for the quantitative distinction of MC-LR was proposed. The assay utilized QD525 and PoPo3 dye as a fluorescence quencher and fluorescent probe. The confinement of the MC-LR target by the aptamer caused changes in the aptamer’s conformation and increased the donor-acceptor distance, resulting in an enhancement in the fluorescence response by fording the FRET phenomenon. Despite its reasonable specificity and sensitivity, the proposed aptasensor has some limitations due to the interference of temperature, pH, and concentration of ions. Furthermore, Liu et al. [30] designed a stable, reliable, highly sensitive, and particular electrochemical aptasensor to quantify MC-LR. A novel ternary composite including AuNPs-molybdenum disulfide (MoS_2_) compound enveloped by TiO_2_ nanobeads (TiONBs) was applied to construct the aptasensor. To set up the nanocomposite, TiONBs were integrated to provide a scaffold for growing MoS_2_ nanosheets, and AuNPs were deposited on the MoS_2_ nanosheets as the binding sites for the aptamer. Due to its porous and lamellate nanostructure and high biocompatibility, this improved three-dimensional nanocomposite (AuNPs-MoS_2_) exposed a substantially higher specific surface area for aptamer loading. The spherical TiONBs expanded the exposing surface region for the biomolecules and their high conductivity improved the electron transfer. The horseradish peroxidase (HRP) was utilized efficiently for signal amplification. Through an Au-S bond, the thiolated aptamer was fixed on the AuNPs of the composite surface. After the introduction of MC-LR and biotin-cDNA (a complementary sequence of the aptamer) to the system, MC-LR bound to the aptamer caused a decrease in the hybridization of the biotin-cDNA with the aptamer. As a consequence, the quantity of avidin-HRP attached to the biotin-cDNA was reduced, which declined the catalysis of the reaction between hydroquinone and H_2_O_2_ to yield benzoquinone. With high precision, selectivity, and stability, the aptasensor is potent for target detection in real samples.

In this regard, Xu, et al. [16] produced an aptasensor, which was a simple, reliable, and sensitive aptasensor based on electrospinning and seeded growth techniques coupled with the aptamers on the metal-MOF of the solid-phase microextraction (SPME) fiber. Stainless steel wires (SSWs) were used as the substrate of SPME coatings. The heterobifunctional ligand was applied to modify the MOF by uncovering the alkyl amine groups on their outer layer. The resultant MOFs possessed exceptionally high outside surface regions and desirable mechanical properties (elasticity and toughness). In the presence of tert-butyl nitrite (t-BuONO) and trimethylsilyl azide (TMSN_3_), the alkyl amine group changed to the azide functionalities, which permitted the incorporation of the dibenzylcyclooctyne (DBCO) functionalized-aptamer on the MOF surface by the click chemistry. The SPME with the LC-MS method clarified the amounts of MC-LR. Although this Apt-SPME fiber required special equipment and at least 72 h for the nanofiber preparation, the developed aptasensor could distinguish the ultra-trace levels of MC-LR in the complex samples compared with the traditional SPE or SPME columns (Figure 1d).

Additionally, Xie et al. [32] established a simple nano-confinement fluorescent aptasensor with the combination of DNase-I enzyme-assisted signal amplification analytical strategy for the ultra-low detection of MC-LR. A nanopipette was applied as a substrate for embedding the aptasensor components. AuNPs, as well-known quenchers and suitable carriers, are effective FRET receptors through adsorbing the FAM-labeled aptamer strands. The FAM-labeled aptamer and AuNPs solution filled in the nanofluidic area of the nanopipette. In the absence of the target, the FAM-labeled aptamers anchored on AuNPs, which manifested in negligible fluorescence emission. In the MC-LR presence, the aptamer bound with MC-LR and left the surface of AuNPs. The aptamer was cleaved by adding DNase-I enzyme, and MC-LR was delivered as an extra analyte particle to accomplish enhance the signal, acquiring a potent fluorescence response. So, this confined nanoscale platform can be aggressive for the ultra-low detection of MC-LR in biological samples.

### 3.2. Saxitoxins (STXs)

Saxitoxins, made by cyanobacteria, can also be found in freshwater. Saxitoxins are heat-stable and water-soluble. They are tasteless and odorless and are not destroyed by cooking. STXs are selective, reversible voltage-gated sodium channels, that work by directly attaching to the voltage-dependent sodium channels in the membranes of nerve and muscle cells, disrupting nerve signaling and causing paralysis [57].

#### A Brief Lookat Aptasensing Stages on Saxitoxins (STXs)

Today, dual-mode sensing tools are efficient for commercial diagnostic assays that facilitate target detection [59,60]. Dual-response sensors produce multicolor output signals, efficient for multiple target detection [61]. In some cases, single-response sensors suffer from low reproducibility due to possible interferences induced by intrinsic and external agents, e.g., the effect of background signals. By contrast, a dual-signal sensor can efficiently conquer the defects of single-response sensors through a built-in intrinsic correction of background signals. In dual-response sensors producing two different output signals, a comparison of the signals confirms the accuracy of both the proposed sensing mechanism and sensor functionality, which eliminates the need for conventional analytical methods for validating the sensor. Hence, dual-response sensors possess improved accuracy and reliability [62]. However, no dual-mode aptasensor for this toxin (STX) has been fabricated based on experiments. Consequently, the creation of dual response aptasensors may open up new possibilities for the accurately detecting of STX. Procedures as mentioned above, different types of aptasensors have been developed for distinguishing marine toxins. The most recent aptasensor designs—a subset of electrochemical, label-free colorimetric, and label-free fluorescent aptasensors—are discussed in this section. In addition, the details and modifications of the most relevant sequences of aptamers that were utilized in these platforms have been described in Table 2.

In this context, Qi et al. [43] fabricated an innovative electrochemical aptasensor for the susceptible and selective detection of STX. Figure 2(aA) displays the design of the electrochemical aptasensor. A DNA nanotetrahedron was designed to stabilize a DNA triplex along with the specific aptamer on the electrodes. As is depicted in Figure 2(aB,aC), after the formation of the aptamer-triplex on the top vertex of the immobilized nanotetrahedron, primary electrochemical signals (Ib) have decreased (Ic). In the presence of the target STX, the aptamer-triplex structure switched and the AptSTX-pyrimidine arms dissociated from the electrode, resulting in a significant signal change from Ib to Id with different lengths of the triplex. The aptasensor with a recovery of 94.4–111% in the real seawater samples showed ultra-sensitivity and selectivity.

As a result of the study conducted by Li et al. [44], a label-free colorimetric aptasensor for STX based on a terminal-fixed anti-STX aptamer (TF-M-30f) to overcome the challenges of structural stability and affinity of the aptamer was constructed. In the absence of STX, the TF-M-30f was hybridized with the cDNA strand. So, there was no free cDNA in the solution to adsorb on the surface of AuNPs for activating their peroxidase-like function. Consequently, AuNPs could not oxidize TMB, and the solution remained colorless. With adding STX, it interacted with the TF-M-30f strand. The released cDNA strands were adsorbed on the surface of AuNPs which activated their oxidation function. The oxidized TMB induced a blue color of the solution that clarified the presence of STX. The binding affinity to STX was boosted by nearly 145 percent when the TF-M-30f aptamer with terminal fixation was used instead of the original aptamer. Besides, Park et al. [45] developed an electrochemical aptasensor with high sensitivity and selectivity to detect STX in real freshwater samples. The porous pPtNPs and STX aptamer were used as the enhancer of the electrochemical signal and bioprobe, respectively. Thiolated STX aptamer interacted with the pPtNPs and formed the pPtNP/STX complex on the electrode surface. The STX interaction with the aptamer on the surface of the pPtNP-modified electrode changed the electrochemical response.

Based on logical site-directed mutation and truncation, Zheng et al. [63] created an optimized aptamer (M-30f) with a greater affinity for STX than the original aptamer. They used a STX-specific aptamer sequence for this approach, which was previously found using the Systematic Evolution of Ligands by Exponential Enrichment (SELEX) technique. Then, some mutations were created based on the secondary structure prediction of the aptamer to increase the binding affinity. Finally, the dispensable nucleotides for binding to STX were truncated to reduce the production cost. As a result, the proposed M-30f aptamer with the K_d_ value of 133 nM possessed a 30-fold higher affinity to STX than the parent aptamer.

Additionally, Cheng et al. [46] developed an aptasensor to detect STX through the conformational switch of the aptamer. The M-30f aptamer with a short sequence (30 bases) and the highest binding affinity toward STX (133 nmol) was used to construct the aptasensor. The fluorophore (HEX) and quencher (BHQ1) molecules were attached at the 3′ and 5′ ends of the aptamer), respectively, to identify the structural switch of the aptamer. The binding of the dye-labeled M-30-f to STX increased the distance between the BHQ1 transmitter and the HEX receptor, which induced a potent fluorescence response. The aptasensor with high sensitivity and selectivity can be a suitable alternative for detecting STX in marine food samples (Figur 2c). As mentioned, a DNA nanotetrahedron was applied by Qi et al. [43] to construct the electrochemical aptasensor for monitoring STX. However, the diverse arrangements of the aptamers can be used for future construction of STX aptasensor, such as triple-helix molecular switch (THMS), three-way junction pocket, hairpin-like, ladder-shape, sandwich-like, arch-shape, infinity-shape, and Y-shape [64,65]. Besides, developing label-free fluorescent aptasensors for monitoring STX instead of labeled ones [46] can efficiently represent simple and facile sensing tools as a future goal.


biosensors-13-00056-t002_Table 2Table 2Detailed information on different MC-LR aptamers applied in the sensors.TargetAptamer ModificationAptamer Sequences5′-3′Length (nt)Kd (nM)Ref
**MC-LR**
5′-SHTTT TTG GGT CCC GGG GTA GGG ATG GGA GGT ATG GAG GGG TCC TTG TTT CCC TCT TG-5550[31]5′-HS-(CH2)6GGC GCC AAA CAG GAC CAC CAT GAC AAT TAC CCA TAC CAC CTC ATT ATG CCC CAT CTC CGC6028 ± 8 nM[30]-GGC GCC AAA CAG GAC CAC CAT GAC AAT TAC CCA TAC CAC CTC ATT ATG CCC CAT CTC CGC6028 ± 8 nM[16]5′-NH2-(CH2) 6TTT TTG GGT CCC GGG GTA GGG ATG GGA GGT ATG GAG GGG TCC TTG TTT CCC TCT TG5650[29]5′-NH2-(CH2) 6GGC GCC AAA CAG GAC CAC CAT GAC AAT TAC CCA TAC CAC CTC ATT ATG CCC CAT CTC CGC6028 ± 8 nM5′-NH2-(CH2) 6CAC GCA ACA ACA CAA CAT GCC CAG CGC CTG GAA CAT ATC CTA TGA GTT AGT CCG CCC ACA60925′-NH2-(CH2) 6CAC GCA CAG AAG ACA CCT ACA GGG CCA GAT CAC AAT CGG TTA GTG AAC TCG TAC GGC GCG601035′-(SH)-(CH2)6-GGC GCC AAA CAG GAC CAC CAT GAC AAT TAC CCA TAC CAC CTC ATT ATG CCC CAT CTC CGC6028 ± 8 nM[33]5′-(SH)-(CH2)6GGC GCC AAA CAG GAC CAC CAT GAC AAT TAC CCA TAC CAC CTC ATT ATG CCC CAT CTC CGC6028 ± 8 nM[32]AAAAAAAAAAAAAGGC GCC AAA CAG GAC CAC CAT GAC AAT TAC CCA TAC CAC CTC ATT ATG CCC CAT CTC CGC6028 ± 8 nM[20]FAMGGC GCC AAA CAG GAC CAC CAT GAC AAT TAC CCA TAC CAC CTC ATT ATG CCC CAT CTC CGC6028 ± 8 nM[35]QD525GGC GCC AAA CAG GAC CAC CAT GAC AAT TAC CCA TAC CAC CTC ATT ATG CCC CAT CTC CGC6028 ± 8 nM[36]-GGC GCC AAA CAG GAC CAC CAT GAC AAT TAC CCA TAC CAC CTC ATT ATG CCC CAT CTC CGC6028 ± 8 nM[37]5′-SH-TTTTTTGGC GCC AAA CAG GAC CAC CAT GAC AAT TAC CCA TAC CAC CTC ATT ATG CCC CAT CTC CGC6028 ± 8 nM[38]
**STX**
5′-SH-CTTCTTCTTCTT&TTCTTCTTC-3′TTG AGG GTC GCA TCC CGT GGA AAC AGG TTC ATT G34133 nM[43]FAMGGC GGG TTT TGA GGG TCG CAT CCC GTG GAA ACA GGT TCA TTG TTC CCG CC5032.8nM[44]5′-ThiolGGT ATT GAG GGT CGC ATC CCG TGG AAA CAT GTT CAT TGG GCG CAC TCC GCT TTC TGT AGA TGG CTC TAA CTC TCC TCT78-(long)[45]HB-M-30f (HEX 5′-fluorophore (HEX, hexachlorofluorescein) & quencher (BHQ1, black-hole quencher 1)-3′TTG AGG GTC GCA TCC CGT GGA AAC AGG TTC ATT G34133 nM[46]


According to research by Ng et al. (2012) and, Table 2, the best and most aptamers with the lowest Kd (28 ± 8 nM) that was employed against MC-LR were selected [15,20,29,30,32,35,36,37,38]. These aptamers feature two or more selectivities for binding to the seven-cyclic peptide structure of MCs congeners at positions 2 and 4 [66]. In addition, the most STX aptamer that was used by Cheng et al. 2018 and Qi et al., 2020 with the lowest Kd (133 Nm), is an appropriately designed aptamer for STX [43,45,46]. In the contrast, the lowest Kd was 32.8 nM with the terminals fixed which has a 145-fold enhancement of binding affinity in the study by Li et al. 2021 [44].

## 4. Monitoring of Phycotoxin Pollutants in Fish

### 4.1. Tetrodotoxins (TTXs)

A potent neurotoxin called tetrodotoxin (TTX) causes several human intoxications and fatalities every year. Although the source of TTX is uncertain, endosymbiotic bacteria that frequently seem to be transferred down the food chain appear to create it, especially in pufferfish. This neurotoxic, which has been connected to health problems in Asian countries, has recently been found in the Pacific and Mediterranean due to global warming. TTX blocks sodium channels, often resulting in heart failure and death because there is no known cure for it. A legal cap of 2 mg equivalent TTX/kg was set in Japan [67]. Some of the apta-platforms created for the detection of TTX are described below following the most recent studies:

#### Some Aptasensing Platforms on Tetrodotoxins (TTXs)

For the first time, Liu et al. [39] created an ultrasensitive AuNPs@MIL-101-based aptasensor to detect TTX via a dual-response mechanism. Figure 3a illustrates the designed aptasensing mechanism. A specific TTX aptamer as the diagnosis element was used along with AuNPs and MIL-101 metal-organic framework (MOF) as the quencher and surface-enhanced Raman spectroscopy (SERS) substrate, respectively. in the absence of TTX, the Cy3-tagged aptamers were adsorbed on the AuNPs@MIL-101 substrate potently. The quenching and plasmonic properties of AuNPs induced a decrease in the fluorescence signal and increased SERS intensity. By adding TTX, the aptamer-TTX complexes formed, which were released from AuNPs@MIL-101. So, the fluorescence signal intensified whereas the SERS 4.14 signal dropped.

Additionally, Lan et al. [40] developed a low-cost and empathetic method with a label-free aptamer to recognize TTX. The fluorescent reporter that intercalates into the DNA strand in this fluorescent aptasensing technology is berberine. In the presence of the target, the aptamer structure altered from a curly form to a neck ring one, which caused a better interaction of berberine into the aptamer-TTX complex. As a result, the fluorescence intensity increased effectively. Givem that berberine has not been employed as a fluorescent reporter for TTX quantification in any platform, this approach can be used for the rapid and low-cost detection of TTX with great sensitivity.

An aptasensor was devised in this respect by Zhang et al. [42] that proposed a novel aptasensor by using magnetic nanoparticles (MNPs) and a triple cycle improvement method (strand displacement amplification (SDA) accompanied by the catalytic hairpin assembly (CHA)) for TTX detection. As indicated in Figure 3b, after the immobilization of the TTX aptamer on MNPs, there was a competition between TTX and cDNA to interact with the specific aptamer. The remained cDNA fragments in the solution were used as a primer for the SDA mechanism. The triple cycle amplification created a large amount of ssDNA. The ssDNA strands disrupted the reporter probe containing a hybrid of the 6-fam-tagged strand and the BHQ-1-tagged one. As a result of establishing a complex of the 6-fam-labeled strands with the ssDNA, the fluorescence response increased, indicating the presence of TTX. The aptasensing method with a LOD as low as 0.265 pg mL and high sensitivity and stability is a reliable strategy for analyzing the trace amount of TTX in food samples.

Considering the present studies on the TTX aptasensing approaches, it is great potential to develop colorimetric aptasensors to detect TTX. By providing the naked-eye monitoring of the target, colorimetric aptasensors eliminate the need for high-cost equipment. Besides, the development of colorimetric aptasensors for TTX represents simple, facile, and easy-to-use detection tools that are promising lab-on-chip devices [39]. AuNPs@MIL-101 was utilized for the SERS detection TTX [39], while there are diverse MOF nanostructures [40] that are advantageous for designing TTX aptasensors. For example, the UiO-66 framework, with the benefits of a simple production process, flexible porous structure, large surface area, and reactivity, is superior in the field of aptasensor design. Although Lan et al. [40] used berberine dye to create a label-free fluorescent aptasensor for TTX, there are other fluorescent intercalating dyes suitable for constructing a simple and cost-effective tag-free aptasensor for monitoring TTX, such as SYBR Green I (SGI), OliGreen, DAPI, PicoGreen, Thiazole orange, and so on [39,40]. SGI is appropriate for commercial tag-free aptasensors for TTX due to its strong thermal stability, great optical characteristics, and minimal fluorescent background [40,42] (Table 3).


biosensors-13-00056-t003_Table 3Table 3Detailed information on different TTX aptamers applied in the sensors.TargetAptamer ModificationAptamer Sequences5′-3′Length (nt)Kd (nM)RefTTX5′-cyanine-3 dyeTCA AAT TTT CGT CTA CTC AAT CTT TCT GTC TTA TC351.20 ± 0.25 μM[39]5′-HS-(CH2)_6_AAA AAT TTC ACA CGG GTG CCT CGG CTG TCC30-[40]5′-biotin-TCA AAT TTT CGT CTA CTC AAT CTT TCT GTC TTA TC351.20 ± 0.25 μM[42]


## 5. Monitoring of Phycotoxin Pollutants in Fish and Shellfishes

### 5.1. Okadaic Acid (OA)

Okadaic acid and structurally related dinophysistoxins (DTXs) are polyketide molecules produced by several dinoflagellates. The LD50 for OA is 0.2–0.225 mg kg^−1^ BW. Similar to MC-LR, okadaic acid performance can inhibit the activities of protein phosphatases type 1(PP1) and type 2A (PP2A) that cause hepatotoxicity that is often lethal to animals and responsible for human diarrhetic shellfish poisoning [68]. People often get severe diarrhea and abdominal pain within 0.5–12 h of consuming shellfish contaminated with okadaic acid or DTXs. According to some concerns, okadaic acid may be neurotoxic, immunotoxic, and embryotoxic which can be a tumor promoter, and tumor promoters raise the risk of getting several types of cancer [69].

#### Recent Development in Aptasensing Platforms on Okadaic Acid (OA)

To rapid and sensitive detection of OA, Rouhbakhsh et al. [47] engineered a novel strategy based on liquid crystals (LCs) for the sensitive detection of OA, as depicted in Figure 4a. As a linker, a short complementary sequence of an OA aptamer was bound on the sensing platform. Along with the particular aptamer, a signal amplifier strand connected to AuNPs (SA-AuNPs) was employed to enhance the light signal of LC pictures. In the absence of OA, the double-stranded structure formed on the sensing platform through hybridizing the two ends of the specific aptamer by the linker and amplifier strands disrupted the alignment of LCs. So, cell pictures grew bright. The presence of OA caused the separation of the duplex sequence, and, as a result, LCs were ordered, achieving the dark LC images. The sensing strategy provided great sensitivity with high selectivity for OA.

LCs, as unique substances with fluidity, optical anisotropy, and long-range rotational regularity, hold promise for ultra-sensitive target monitoring. The reorientation behavior of LCs appears within a few seconds, which facilitates obtaining reinforced optical signals from biomolecular events [64]. Thus, applying the different aptamer structures (e.g., three-way junction pocket, THMS, π-shape, and so on) accompanying with LCs achieves high-throughput sensing tools for OA. Rouhbakhsh et al. [47] applied AuNPs to develop an LC-based aptasensor for monitoring OA. At the same time, there is a wide range of nanomaterials that can be advantageous for developing LC-based aptasensors, such as carbon-based nanomaterials, metallic nanostructures, MOFs, QDs, MXene, and so on.

In this regard, Shan et al. [48] suggested a straightforward and quick method for detecting OA in seafood. The OA-specific aptamer was designed as a duplex (aptamer-5T and cDNA) and fixed on the surface of streptavidin magnetic beads (SMBs) through the cDNA strands. The aptamer-5T was detached from the cDNA and moved away from the SMB surface in the presence of OA. Following the magnetic separation, the terminal deoxynucleotidyl transferase (TDT) enzyme and 2′-deoxythymidine 5′-triphosphate (dTTP) were added to the supernatant to generate the fluorescent CuNPs using the long ploy(thymine) strand at the end of the cDNA as a scaffold. So, an increase in the fluorescence response clarified the OA presence in the sample.

Wang et al. [7] introduced the aptamer-based microcantilever array, which provides a quick and sensitive assessment of OA in marine animals. As shown in Figure 4b, the thiol-modified aptamer was first immobilized on the microcantilever. The binding of OA to the aptamer strands led to changes in the electrostatic interactions and hydrogen bonding. The mechanical signals created by these interactions increased the surface tension of the gold side of the microcantilever, causing the microcantilever to deviate. The deflection of the emitted laser beam to the surface of the microcantilever was used to calculate the deviation. The proposed array detects OA in a cost-effective, label-free, and fast manner.

As a result of the study conducted by Ramalingam et al. [70], an electrochemical aptasensor was designed by using a PDMS-based integrated SPCE microfluidic chip. The 2D nanocomposite of black phosphorus (BP) and AuNPs were used to modify the output signal and boost the stability of the aptamer on the biochip. The binding of the particular aptamer to OA altered charge transport on the electrode surface and hence the electrochemical response of the aptasensor. The aptasensor, with a reaction time of 30 min and a detection limit of 8 pM, may be utilized as a sensitive and simple practical assay for biochemical assays (Figure 4c).

The only piezoelectric aptasensor designed by Tian et al. [49] is a piezoelectric aptasensor based on the piezoelectric effect of quartz crystal. AuNPs-modified reporter DNA was used to improve the detection performance and amplify the signal. A Quartz crystal microbalance (QCM) device was applied as a transducer to identify the produced signal. Through Au-S bonding, the thiol-modified capture DNA adhered to the gold surface of the QCM device. The capture DNA hybridized the specified aptamer at one end and immobilized the reporter DNA on the QCM at the other end. The higher the amount of OA induced, the less aptamer on the QCM surface; therefore, a significant number of the reporter strands were released from the QCM surface, which changed the frequency signal. Compared to the previously reported biosensors, the aptasensor can be used as a promising method for the ultra-fast, sensitive, and cheap diagnosis of OA. It can also be helpful in other fields, such as the food industry and water quality control.

Furthermore, Zhao et al. [50] presented a promising aptasensing strategy based on a SERS approach for OA detection. The aptasensor was composed of two composite layers, the primary layer of AuNPs and the second layer of three-dimensional succulent-like silver particles (3D-SS), to create a larger surface and high-density hot spot. It is also composed of silver nano octagons connected to the signal probe (Cy3-DNA1-SNO) to amplify the response signal. First, the Au substrate was treated by 3D-SS. Then, the aptamer strands were attached to the Au substrate and hybridized by Cy3-DNA1-SNO strands. In the presence of OA, it combined with the aptamer caused the release of the Cy3-DNA1-SNO from the Au substrate and produced a SERS signal. The aptasensor is a promising, ultra-sensitive, rapid, and low-cost tool for detecting environmental OA.

Enzyme-based signal amplification techniques are one of the improvement approaches for the sensitivity of aptasensors. The strategies act through enzymatic recycling of the target. Enzyme-assisted amplification methods with great catalytic efficiency provide commercial detection tools with excellent sensitivity. Various endonucleases and exonucleases have been applied in the target recycling strategy. However, just TDT enzyme [48] was utilized for OA detection. Hence, developing enzymatic aptasensing arrays can be promising for monitoring OA.

### 5.2. Brevetoxins (BTXs)

Brevetoxins (BTXs) are lipid-soluble cyclic polyether marine compounds that produce neurotoxic shellfish poisoning in humans. This syndrome is called neurotoxic shellfish poisoning (NSP) [71]. Consuming contaminated food and direct inhalation of contaminated water aerosols are sources of exposure [72]. These compounds can bind to voltage-gated sodium channels in nerve cells and disrupt normal neurological processes, then alter the membrane properties of excitable cell types in ways. BTXs, through enhancing the inward flow of Na^+^ ions into the cell, cause neuron depolarization [73]. The brevetoxins are tasteless, odorless, and heat and acid-stable, which can be murderous to fishes, birds, and mammals with (lethal dose resulting in 50% deaths) LD_50_ = 0.20 mg/kg body weight. In human cases, more than 18 μg brevetoxin in each mg of clams has been reported, causing neurotoxic shellfish poisoning. Given that these toxins cannot be easily removed by food preparation procedures, the best solution is to detect them before consumption [74].

#### Aptasensing Platforms on Brevetoxins (BTXs)

Brevetoxin can be assayed by using various methods like ELISA, antibody radioimmunoassay, and so on. Nowadays, aptasensores are known as the newest detection methods. Herein, we have reviewed the latest aptasensing platforms on brevetoxins. Recently, Essia et al. [75] developed a competitive electrochemical aptasensor merged by Faradaic impedance spectroscopy (EIS) and a specific aptamer for monitoring BTX-2. BTX-2 was immobilized on the surface of the cysteamine-treated Au electrode by attaching it to a 1,4-phenylene diisocyanate (PDIC) linker by its hydroxyl group. In addition to the aptamer strand, it was complexed with the immobilized BTX-2 on the electrode surface that changed the R_CT_ of the [Fe(CN)_6_]^4−/3−^ redox agent. The aptasensor provided the susceptible detection of BTX-2 with no cross-reactivity via two similar congeners, BTX-3, and other marine toxins. Moreover, a good recovery percentage was obtained in the spiked shellfish extracts. Thus, in terms of simplicity, stability, tag-free, and low cost, the constructed aptasensor is competitive with the other commercial approaches for monitoring BTX-2 in shellfish.

In this context, Cagalyan et al. [52] designed a novel optical aptasensor for BTX detection by using two methods consisting of spectroscopic ellipsometry (SE) and attenuated internal reflection spectroscopic ellipsometry (TIRE), running under surface plasmon resonance (SPR). As a result of the high sensitivity of ellipsometry to molecular deposition on the surface, the phase shift (Δ) in the polarization states of the reflected light can be measured for detecting the deposited structures on the surface. Hence, the amine-modified aptamers were immobilized on the mercapto-propyl trimethoxysilane (MPTES)-modified Si-wafer pieces. The Δ parameter significantly changed during the target recognition based on its complexation with the attached aptamers on the wafer surface. As a shortage, the selectivity of the aptasensor was weak at the high concentrations of the interfering compounds, especially OA with structural similarity to BTX. The aptasensor was utilized effectively to identify BTX components spiked into genuine fish and shrimp tests.

Considering the present studies on the BTX aptasensing approaches, Ramalingam et al. [9] reported a nine MHz AT-cut quartz crystal resonator modified by the specific single-stranded DNA aptamer for detecting BTX. In the detection array, an acoustic wave sensor was used, in which the electrode surface was modified by the thiolated aptamer. In the presence of BTX molecules, the 3D conformation of the aptamer was altered by capturing the target. So, the change in the crystal frequency was proportional to the BTX concentrations and changes in the mass deposited on the crystal surface. The limit of detection was obtained at 220 nM which was lower than CODEX threshold limits (0.8 mg/kg = 922.6 nM). The overnight time was taken for the aptamer fixation; the pattern procurement was achieved at 45 min. Relying on the reusability and stability of the aptasensor over multiple runs can be beneficial in a genuine factory.

Based on the studies, optical, electrochemical, and acoustic wave biosensors were utilized for BTX detection. There is excellent potential to develop microfluidic arrays as portable tools for the on-site and real-time detection of BTX. There are various substrates for constructing easy-to-carry aptasensing tools, including paper, nanofibers, glass slides, plastic sheets, micropipette tips, etc. [21,76] Combining colorimetric strategies with microfluidic arrays can achieve cost-effective and simple BTX detection just by the naked eye.

### 5.3. Aptasensing Platforms for the Other Phycotoxins in Fish and Shellfish

#### 5.3.1. Designed Aptasensing Platforms on Gonyautoxin (GTX)

GTX1/4 is one of the most potent marine toxins belonging to paralytic shellfish poisoning (PSP). GTX1/4 has caused worldwide concern as a biological risk currently; no effective treatment is known for it, and the only way to deal with it is through preventive diagnosis. G018-T-d is a truncated ssDNA aptamer that recognizes GTX1/4 at nanomolar levels but with low affinity. Song et al. [77] applied de novo computational approaches after SELEX to optimize the G018-T-d strand with high affinity. By predicting the 3D model and nucleotide sequence of the aptamer-based on circular dichroism measurements, it was found that at the 5′ end of G018-T-d, there is a barrier for the movement of GTX1/4 towards the aptamer binding pocket. However, the 5′ end has the characteristics of a G-quadruplex structure, which has a high binding affinity to the target analog. Sequentially simulating spontaneous binding clarified that by removing the first five nucleotides at the 5′ end of G018-T-D, the binding pocket became available, which led to a 20-fold increase in the binding affinity of the aptamer (tG018-T-d). So, tG018-T-d can be used as a promising engineered aptamer for GTX 1/4 to develop highly efficient sensing kits.

In this context, Gao et al. [25] proposed a high-affinity engineered aptasensor for GTX1/4 based on a label-free biolayer interferometry (BLI) platform. The G018-T-d aptamer was shortened based on secondary structure and developed by the GO-SELEX technique (K_d_ = 17.7 nM) to increase its binding affinity to GTX1/4. The in vitro-selected aptamers were fixed on the surface of the BLI platform. The thickness of the biological layer increased by interacting with GTX1/4 which changed the output signal. The aptasensor is promising for the optical detection of GTX1/4 in seafood samples with high stability and reproducibility. Given that GTX1/4 is a significant marine toxin with severe hazards to human health, it is essential to expand aptasensing approaches for sensitive determination of GTX1/4. Different aptasensors can be developed to detect GTX1/4 based on the colorimetric, fluorescent, electrochemical, luminescent, and field-effective transistor techniques using diverse nanosheets, nanospheres, nanocomposites, core-shell nanostructures, and so on.

#### 5.3.2. A Developed Aptasensor for the Identity of Palytoxin (PTX)

Palytoxin (PTX) by LD50 can build up at extremely high levels in shellfish and crabs throughout the aquatic food chain, and eating seafood contaminated with PTX injures humans and results in symptoms, including weakness, disorientation, muscle pain, breathing problems, heart failure, and even death. A 30 mg/kg maximum regulatory limit has been suggested by the European Food Safety Authority for clam flesh. Analytical techniques, however time-consuming and requiring expensive equipment, are not appropriate for on-site monitoring and fall short of the regulatory standard for PTX (30 mg/kg) in shellfish. This has led to the development of various biosensing technologies like aptasensore. For example, Gao et al. [24] designed a sensing strategy for the rapid detection of deficient amounts of PTX.13 in food sources by using the BLI platform as a real-time optical analytical technique. In this system, analytes that interact with ligands immobilized on a sensor surface form a monomolecular layer, which causes a proportional shift in the interference spectrum of the reflected light, it can detect changes in the concentration of targets. First, the aptamer was selected by the MB-SELEX technique and the horseradish peroxide (HRP) enzyme-labeled aptamer interacted with PTX on the PTX-modified BLI surface. Then, the BLI platform containing the PTX: HRP-aptamer complex was submerged in a 3,3′-diaminobenzidine (DAB) solution, resulting in a precipitated polymeric product on the aptasensor surface. So, the optical thickness and mass density of the aptasensor layer changed considerably. It caused changes in the reflected light, which could be used to measure the concentration of the target. The designed platform is capable to detecting PTX in less than 10 min with a detection limit of 0.04 pg/mL, promising for monitoring PTX in food. With the aid of HRP, the sensitive detection of PTX was achieved successfully. However, there are DNA or RNA single strands with specific catalytic activity, known as aptazymes (DNAzyme/RNAzyme), that can catalyze the biological reactions of oligonucleotide cleavage, phosphorylation, ligation, and porphyrin metalation [78]. The use of aptazymes for developing sensing methods eliminates the need for enzymes that simplifies detection assays. So, introducing SELEX-extracted aptazyme for PTX and designing aptazyme-based biosensors is promising for future studies.

#### 5.3.3. A platform of Anatoxin-a (ATXs) Aptasensing

Anatoxins (ATXs) are alkaloid toxins that are highly polar and fully soluble in water. This compound is the most fully natural neurotoxin produced by freshwater cyanobacteria with an LD_50_ = 20 μg kg^−1^ [79]. This toxin is a potent agonist acetylcholinesterase which binds to nicotinic acetylcholine receptors and induces conformational changes in the postsynaptic receptor ion channel complex. It can act like an organophosphorus and carbamate insecticide. Organs such as the liver, kidney, lung, thymus, spleen, adrenal glands, intestinal tract, immune system, and heart that have several acetylcholine receptors can be hurt after toxin exposition. The side effects of human toxicity include a total loss of coordination, paralysis in the skeleton and respiratory muscles, and even death [80]. Anatoxin-a can be assayed by its inhibition of acetylcholinesterase. Since anatoxins have multiple analogs, it is tough to distinguish them. Hence, in recent decades, different methods have been implemented in a biosensor using aptamer for detecting them. Elshafey et al. [27] reported a simple label-free impedimetric aptasensor for ATX based on the conformational change of the aptamer. It was utilized for toxin identification by recording the electron transfer resistance. The surface of the Au electrode was modified by a cysteamine monolayer to hinder the access of the redox marker to the electrode surface. The specific aptamer was located on the modified Au electrode. In the absence of ATX toxin, the negatively-charged [Fe(CN)_6_]^4−/3−^ redox probe was repelled from the electrode surface, which hindered the redox reaction. With the addition of the toxin, the aptamer switched into a different structure that increased accessibility of the [Fe(CN)_6_]^4−/3−^ marker to the surface, consequently decreasing the resistance to the electron transfer. So, an aptasensor with good stability (for 15 days), high accuracy, and great sensitivity has the potential to detect ATX in real samples just in 60 min. Although the aptasensor is stable, fast, and sensitive for monitoring ATX, the development other types of aptasensors facilitate its naked-eye detection. As highly efficient sensing tools, wearable aptasensors provide healthcare monitoring. By embedding in wearable objects, such as smart watches, rings, clothes, glasses, eye-contact lenses, bandages, and so on, wearable biosensors are easy-to-use sensing tools [81]. In the case of ATX, wearable aptasensors can be potent and worthy of development. In addition, Table 4 is useable to find which aptamer is the best and most effective for identifying various phycotoxins (based on the lowest (kd) value).


biosensors-13-00056-t004_Table 4Table 4Detailed information of different aptamers applied in the sensors.TargetAptamer ModificationAptamer Sequences5′-3′Length (nt)Kd (nM)Ref
**ATXa**
5′-HO–(CH2)6–S–S–(CH2)6TGG CGA CAA GAA GAC GTA CAA ACA CGC ACC AGG CCG GAG TGG AGT ATT CTG AGG TCG G5881.3 ± 8 nM[27]
**OA**
-GGT CAC CAA CAA CAG GGA GCG CTA CGC GAA GGG TCA ATG TGA CGT CAT GCG GAT GTG TGG 6023 ± 1.52 nM[47]3′-Biotin3′-FAMATT TGA CCA TGT CGA GGG AGA CGC GCA GTC GCT ACC ACC T40-[48]5′-Thiol-GGT CAC CAA CAA CAG GGA GCG CTA CGC GAA GGG TCA ATG TGA CGT CAT GCG GAT GTG TGG6023 ± 1.52 nM[70]-GGT CAC CAA CAA CAG GGA GCG CTA CGC GAA GGG TCA ATG TGA CGT CAT GCG GAT GTG TGG6023 ± 1.52 nM[49]5′-(SH)-(CH2)6-GGT CAC CAA CAA CAG GGA GCG CTA CGC GAA GGG TCA ATG TGA CGT CAT GCG GAT GTG TGG6023 ± 1.52 nM[7]5′-(SH)-(CH2)63′-BiotinGGT CAC CAA CAA CAG GGA GCG CTA CGC GAA GGG TCA ATG TGA CGT CAT GCG GAT GTG TGG6023 ± 1.52 nM5′-(SH)-(CH2)6-GGC CGC GAG AGA GAC AAC AAG GAT ATA TAT TAT ATG TCG GTT GTA GTG TTG GGT TGC G5892 nM5′-SH-(CH2)6-GGT CAC CAA CAA CAG GGA GCG CTA CGC GAA GGG TCA ATG TGA CGT CAT GCG GAT GTG TGG6023 ± 1.52 nM[50]
**GTX**
-AAC CTT TGG TCG GGC AAG GTA GGT T2517.7nM[25]
**BTX**
5′-SH-(CH2)6GGC CAC CAA ACC ACA CCG TCG CAA CCG CGA GAA CCG AAG TAG TGA TCA TGT CCC TGC G5892 nM[51]5′-ThioMC6-DAT ACC AGC TTA TTC AAT TGG CCA CCA AAC CAC ACC GTC GCA ACC GCG AGA ACC GAA GTA GTG ATC ATG TCC CTG CGT GAG ATA GTA AGT GCA ATC T96-[9]5′-SH–(CH2)6GTG CGT CCC TGT ACT AGT GAT GAA GCC AAG AGC GCC AAC GCT GCC ACA CCA AAC CAC CGG6042 nM[52]5′-NH2-(CH2)6GTT GCC GTC TCC TTA TCC CAC CAC TGC CGA CAC CAC CCC CCC GCG AGA GCG AGA GAG CAC T6196 nM


## 6. Discussion and Conclusions

It is substantial to develop portable sensing tools to monitor phycotoxins to promote food safety, food quality control, environmental monitoring, and human health. For this purpose, representing rapid, uncomplicated, cost-effective, non-invasive, and user-friendly aptasensors is urgent. Hence, we comprehensively reviewed the progress in the aptasensing approaches advantageous for the real-time and on-site determination of phycotoxins. Table 1 clarifies that the aptasensors are superior in food safety and environmental monitoring by providing widespread detection ranges and ultra-low detection limits for the toxins. Besides, high sensitivity, facile application, and rapid response without dependency on skilled operators and intricate equipment make aptasensors unparalleled in detecting phycotoxins. The represented aptasensors can monitor the phycotoxins about picomolar concentration levels (Table 1). However, there are some versatile amplification methods, including hybridization chain reaction (HCR), rolling circle amplification (RCA), recombinase polymerase amplification (RPA), and polymerase chain reaction (PCR), that can improve the sensitivity of the aptasensors. From a future perspective, it is novel to represent ultra-sensitive aptasensors for the toxins with the aid of RCA, RPA, HCR, and PCR amplification methods. Relying on the superiority of the various nanomaterials, nano-aptasensors can quantify the ultra-low levels of phycotoxins. Different nano matters have been applied to construct the aptasensors, such as AuNPs, MoS_2_, PtNPs, MNPs, CuNPs, and MIL-101. While there is a wide range of nanomaterials with high surface area, biocompatibility, and tunable size that can be advantageous to construct efficient aptasensors, such as UiO-MOF, ZIF-MOF, MXene, CNTs, GO, and so on. The unique optical properties of LCs and their intermediate state between solid and liquid phases make them superior aptasensor designs [64]. Table 1 indicates that, up to now, only one LC-based aptasensor has been developed for phycotoxin detection. Relying on the ultra-high sensitivity of the LC-based aptasensors, there is an excellent opportunity to expand this type of sensor for quantifying the toxins in an alone state or in combination with other counterparts. Table 1 indicates that recent studies have focused on developing aptasensing strategies to detect only one type of phycotoxins. From a future perspective, it is crucial to advance the aptasensing mechanisms for the simultaneous monitoring of two or more phycotoxins.

## Figures and Tables

**Figure 1 biosensors-13-00056-f001:**
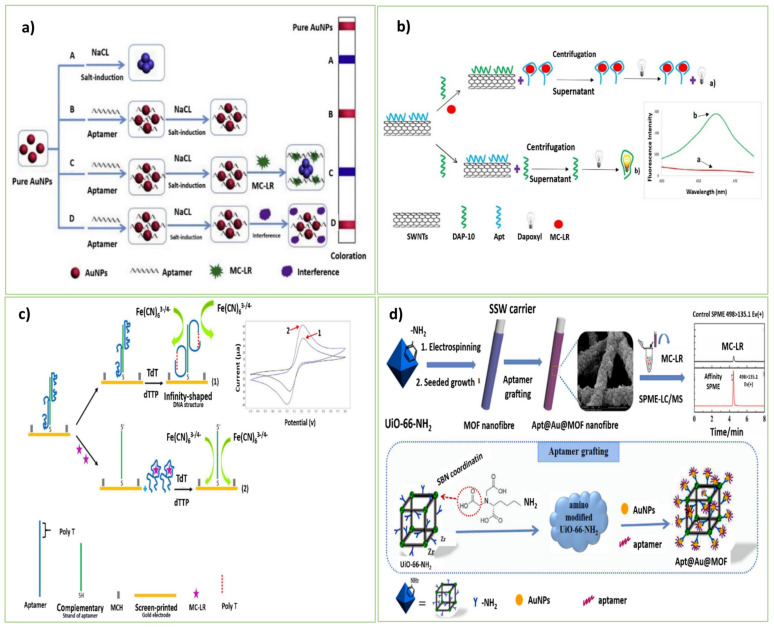
(**a**) Schematic demonstration of the colorimetric aptasensor based on the salt-triggered aggregation of AuNPs. Reprinted by permission from [34]. Copyright 2022, Elsevier. (**b**) Sensing scheme of the electrochemical aptasensor for the MC-LR detection by using the Infinity-shaped DNA structure and TdT enzyme. Reprinted with permission from [35]. Copyright 2022, Elsevier. (**c**) Schematic demonstration of the preparation of a novel affinity Apt@AuNPs@MOF fiber on the SSW, and its SPME procedure of MC-LR [20]. Copyright 2022, Elsevier. (**d**) Schematic description of MC-LR determination based on a fluorescent approach, binding to the surface of SWNTs, resulting in changes in fluorescence intensity [16]. Copyright 2022, Elsevier.

**Figure 2 biosensors-13-00056-f002:**
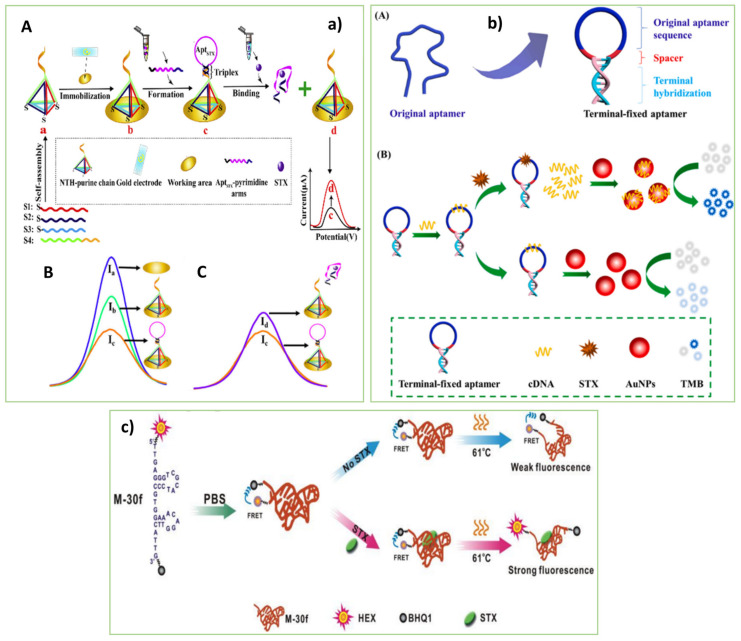
(**a**) Schematic illustration of the electrochemical aptasensor based on nanotetrahedron and aptamer-triplex structures for the detection of STX. Reprinted with permission from [43]. Copyright 2022, Elsevier. (**b**) (**A**) Schematic illustration of the design of the terminal-fixed aptamer. (**B**) Scheme of the label-free colorimetric aptasensor for detection STX [44]. Copyright 2022, Elsevier. (**c**) Schematic illustration of the fluorescence aptasensore (turn on) for STX detection [46]. Copyright 2022, Elsevier. (Capital letters in each part of the images indicate the steps of the process.)

**Figure 3 biosensors-13-00056-f003:**
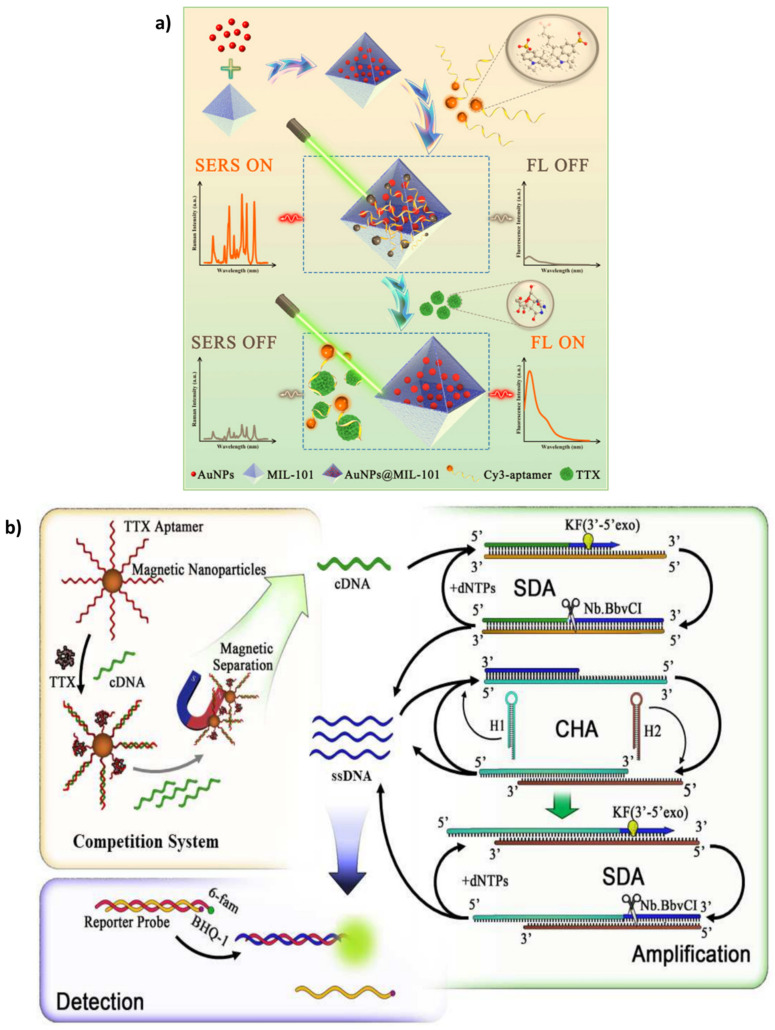
(**a**) Scheme representation of the dual-mode aptasensor for monitoring TTX based on the fluorescence and SERS of AuNPs@MIL-101 nanohybrid. Reprinted with permission from [39]. Copyright 2022, Elsevier. (**b**) The principle of the MNPs-aptamer sensing system under a triple cycle amplification process for the detection of TTX. Reprinted with permission from [42]. Copyright 2022, Elsevier.

**Figure 4 biosensors-13-00056-f004:**
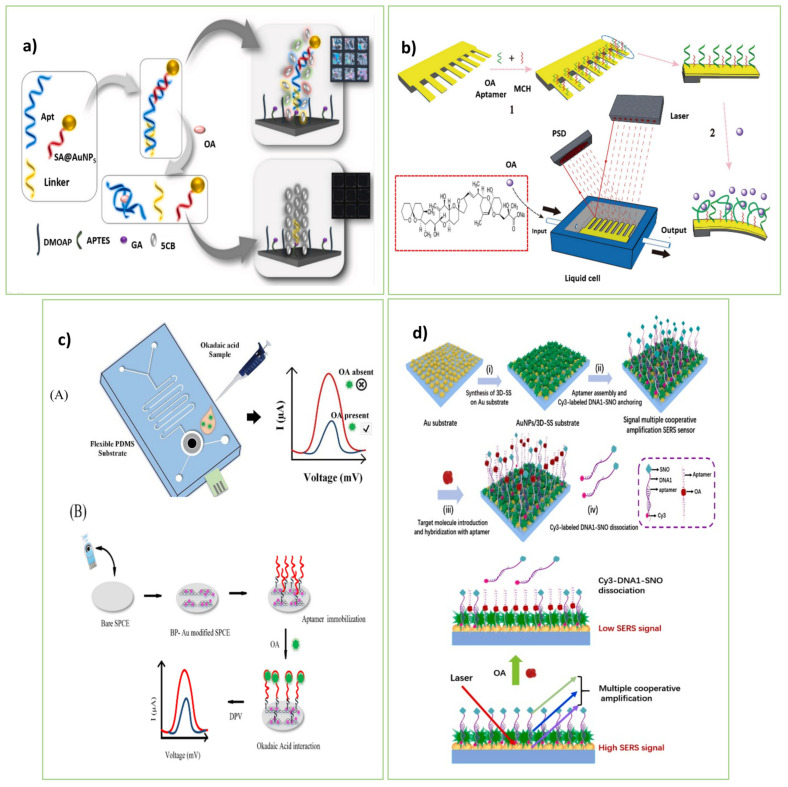
(**a**) Sensing mechanism of the LC-based aptasensor for detecting OA by using AuNP-tagged ssDNA strand. Reprinted with permission from [47]. Copyright 2022, Elsevier. (**b**) Illustration of the aptasensing microcantilever array for the OA detection based on the functionalizing of the array by the thiol-modified specific aptamer. Reprinted with permission from [7]. Copyright 2022, Elsevier. (**c**) Microfluidic electrochemical aptasensor for the detection of Okadaic Acid: (**A**) Graphical of the fabricated PDMS microfluidic chip (**B**) Schematic representation of the process of aptamer-based sensing [70]. Copyright 2022, Elsevier. (**d**) Illustration of the establishment of signal multiple cooperative amplification paper SERS aptasensor and the quantization process of OA. Illustration of the multistage enhancement system of AuNPs/3D-SS composite double layers coupling with SNO [50]. Copyright 2022, Elsevier.

**Table 1 biosensors-13-00056-t001:** Summary of the available aptasensors for the detection of the various phycotoxins.

Target	Method	LOD	Linear Range	Analytical Signal vs. Concentration Slope	Real Sample	Characteristics	Ref
**MC-LR**	AAIA	0.3 μg/L	0.5–4.0 μg/L	*Y* = 9.2288*x* + 8.1072 *R*^2^ = 0.968	Drinking water	A portable analyzer, assay time: 35 min	[29]
	Infinity-shaped DNA structure and TdT enzyme	20 pM in tap water, 35 pM in serum	70 pM–900 nM for tap water and 100 pM–750 nM in serum	*Y* = 1.181*x* − 2.0437 *R*^2^ = 0.99924	Tap water & serum	Assay time: 90 min plus 12 h preparation phase, using DPV analysis	[20]
	Electrospinning & seeded growth on MOF of the solid-phase microextraction (SPME) fiber	0.003 ng/mL	0.008–1.000 ng/mL	*Y* = 12.135665*x* − 33.418 *R*^2^ = 0.9996	Drinking water	Assay time of 20 min plus 72 h preparation phase	[16]
	Dual signal amplification system: HRP enzyme & electroactive nanomaterials	0.002 nM	0.005–30 nM	*Y* = 6.28*logx* + 16.39 *R*^2^ = 0.994	Water from taps, reservoirs & rivers	About 40 h for the synthesis of an AuNP@ MoS2-TiONB nanocomposite and 2 h for the construction	[30]
	Electrochemical impedance spectroscopy (EIS)	1.8 × 10^−11^ mol/L	1.0 × 10^−7^–5 × 10^−11^ mol/L	*Y* = 9.9724*logx* + 111.24 *R*^2^ = 0.997	Water	About 6 h for assay time and preparation phase	[31]
	Signal amplification strategy with DNase I	0.22 nM	0.25–20 nM	*Y* = 0.55*x* − 3.630 *R*^2^ = 0.9996	Drinking water	Directly analysis	[32]
	Cantilever array	1 μg/L	1–50 μg/L	*Y* = 2.11*x* + 9.01 *R*^2^ = 0.97	Buffer	Label-free analysis, About 85 min for assay time and preparation phase	[33]
	Gold nanoparticles (AuNPs) and plasma resonance	0.37 nM	0.5 nM–7.5 μM	*Y* = 0.1662*logx* + 0.2533 *R*^2^ = 0.997	Water	Label-free analyzer, quick detecting: 30 min	[34]
	Graphene-modified screen-printed carbon electrodes (SPEs)	1.9 pM in buffer 1.67 pM inthe spiked sample	0.1 pM–1.0 nM	*Y* = 12.20*logx* + 28.35 *R*^2^ = 0.988	Buffer	Label-free, a stable assay	[14]
	Single-walled carbon nanotubes (SWNTs) & dapoxyl dye	138 pM (0.137 µg/L)	0.4–1200 nM.	*Y* = 0.0089*x* − 2.76 *R*^2^ = 0.9929	Tap water & serum	Label-free analyzer only 75 min for assay time	[35]
	Fluorescence resonance energy transfer (FRET)-based quantum dot (QD)	10^−4^ μg/L	10^−4^–10^2^ ng/mL	*Y* = 0.15*logx* + 0.93 *R*^2^ = 0.94	Eutrophic water	Short lifetime unsuitable for on-site analysis of the target	[36]
	Oriented formation of gold nanoparticle (AuNP) dimers	0.05 nM	0.1–250 nM	$Y=0.12963-043052/1+e^{\left(x+3.736\times 10^{-8}\right)}/4.00852\times 10^{-8}$ *R*^2^ = 0.990119	Water	Achieving results within 5 min	[37]
	Solid-state nanopores	1 μg/L	0.1 nM–20 μM	denotes	Water	-	[38]
**TTX**	SERS	0.006 ng/mL	0.01–300 ng/mL	*Y* = 1470.04*x* + 3386.77 *R*^2^ = 0.9958	Pufferfish and clam meat	Without cumbersome procedures, exhibited signal responses within 1 month, immobilization-free, dual-mode detection	[39]
	Fluorescence reporter	0.074 nM	0.1–500 nM	*Y* = 0.002*x* + 2.4713 *R*^2^ = 0.9958	denotes	Label-free direct analysis, environmental and eco-friendly	[40]
	Docking and molecular dynamics (MD) simulations and microscale thermophoresis (MST)	denotes	denotes	denotes	Pufferfish	Effective repurposing approach, susceptible	[41]
	Triple cycle amplification-based MNPs-apt	0.265 pg/mL	0.05–500 ng/mL	*Y* = 4936.74 + 1327*logx* *R*^2^ = 0.9932	Clams and shellfish	Isothermal amplification, reliable sensitivity, and stability, practical for the analysis of food	[42]
**STX**	Electrochemical aptasensor	0.92 nM	1–400 nM	*Y* = 26.7*x* + 5.48 *R*^2^ = 0.9932	Seawater	Label-free direct analysis, empathetic detection, good practical adaptability and robustness, good recovery, assay time within 30 min	[43]
	Colorimetric aptasensor	0.1423 nM	0.1457–37.30 nM	*Y* = 42.09*logx* + 38.70 *R*^2^ = 0.9863	Seawater and Scallop	Label-free direct analysis, a terminal-fixed anti-STX aptamer, assay time of 75 min, high selectivity, and good recoveries	[44]
	Square Wave Voltammetry (SWV), Electrochemical Impedance Spectroscopy (EIS)	4.669 pg/mL	10 pg/mL–1 μg/mL	*Y* = 0.05282*x* + 0.03536 *R*^2^ = 0.9713	Freshwater	High sensitivity, wide detection range, reduce the error rate	[45]
	Fluorescent aptasensor	1.8 ng/mL	0–24 ng/mL	*Y* = 5.25*x* + 587.2 *R*^2^ = 0.998	Shellfish	Assay time of 30 min, simple	[46]
**ATX**	Impedimetric	0.5 nM	1–100 nM	denotes	Drinking water	Label-free direct analysis, assay time 60 min	[27]
**OA**	LC-based aptasensor	0.42 pM	0.1–100 pM	*Y* = 22.5*x* + 68.6 *R*^2^ = 0.988	Clam	Label-free direct analysis, low-cost detection, rapid	[47]
	Fluorescent aptasensor	1.1 ng/L	1.0 ng/L–50.0 μg/L	*Y* = 438.3*logx* + 3344 *R*^2^ = 0.9909	Shellfish	Ultrahigh-sensitivity, a duplexed aptamer-isothermal amplification, used for on-site food safety screening	[48]
	Aptamer-based microcantilever-array	1 pg/ml	1–5000 pg/ml	*Y* = 29.2496 × 0.4014/(1 + 0.0892 × 0.4014) *R*^2^ = 0.9887	Clam	Label-free direct analysis, excellent dynamic range, economical	[7]
	Microfluidic based aptasensor	8 pM	10–250 nM	*Y* = −0.0142*x* + 6.1139 *R*^2^ = 0.9887	Mussel	Assay time of 30 min, stable, easy to use	[9]
	Piezoelectric aptasensor	0.32 nM	0.5–200 nM	*Y* = 139.0*x* + 489.3 *R*^2^= 0.9826	Mussel	Label-free direct analysis, ultrasensitive, low-cost	[49]
	Paper SERS aptasensor	0.31 ng/mL	1.0–2500 ng/mL	*Y* = 475.55*logx* + 356.20 *R*^2^ = 0.9865	Shellfish	Rapid on-site analysis, low-cost	[50]
**GTX**	Optical BLI aptasensor	50 pg/mL	0.2–90 ng/mL	*Y* = 0.0049*x* + 0.0146 *R*^2^ = 0.998	Shellfish	Label-free direct analysis, stable, good reproducibility, high affinity to GTX1/4, simple detection	[25]
**PTX**	Competitive BLI aptasensor	0.04 pg/mL	200–700 pg/mL	*Y* = (25.05427 − 0.10812)/[1 + (*x*/0.00180)^−1.50701^] + 0.10812 *R*^2^=0.999	Shellfish and seawater	Assay time of 30 min, ultra-sensitive, real-time	[24]
**BTX**	EIS	106 pg/mL	0.01–2000 ng/mL	*Y* = 12.2 + (102.35 − 12.2)/(1 + (*x*/6.66)0.59) *R*^2^ = 0.997	Shellfish	Competitive assay	[51]
	QCR	220 nM/mL	1–1000 nM	*Y* = 0.0752*x* + 12.34 *R*^2^ = 0.997	Shellfish	Label-free direct analysis, assay time of 60 min plus preparation phase	[9]
	SE	720 pg/mL in buffer, LOQ~900 pg/mL in real seafood samples	0.5–2000 nM	*Y* = 3.023*logx* + 2.059 *R*^2^ = 0.95	Fish and shrimp	Label-free direct analysis, assay time of 90 min plus preparation phase	[52]
	TIRE	1.32 ng/mL in buffer, LOQ~1.8 ng/mL in real seafood samples					

## Data Availability

Not applicable.

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
