# Peer review of "Recent Advances in Aptasensing Strategies for Monitoring Phycotoxins: Promising for Food Safety"

_biosensors, 2022, doi:10.3390/bios13010056_

Round 1
Reviewer 1 Report (Previous Reviewer 2)
The author has answered my question and the manuscript can be published.
Author Response
Reviewer #1 comments:
The author has answered my question and the manuscript can be published.
Response: Authors are grateful for your kind point of view on this manuscript.

Reviewer 2 Report (New Reviewer)
Due to the hazard of phycotoxins or marine toxins to humans, livestock, and pets, it is of significance to develop sensitive, specific and fast methods to detect them. Aptamer can identify specific target and is easy to synthesized, so it is commonly employed to establish assays for the detection of its target. This article reviews the designed aptasensors based on the different strategies for the detection of the various phycotoxins. The review is very comprehensive and will provide valuable support for the relevant research. Therefore, I recommend to publish this article after the following issues are addressed.
1. page 8, line 241, the authors cite ref.33 and show the method in Figure 1a. However, in the caption of Figure 1a,"Reprinted with permission from [55]." is used. This needs to be corrected, and I suggest the authors check all the similar situations in the whole manuscript.
2. Page11 line371 “However, any dual-mode aptasensor has not been fabricated for STX. So, developing dual-response aptasensors is novel for the sensitive detection of STX.” I wonder why there is no dual-mode aptasensor used for the detection of STX, is there any problem to solve?
3. Figure 1 and Figure 4 are not very clear, and the words in these figures cannot be well discriminated.
4. Some typos, such as page7 line170 "dditionally"; page9 line284 " Figure 1c)"; page13 line487 “Fishe”; page16 line584 “Fishe”; page19 line685 “humans This”; page20 line741 “Aptasensing platforms on for the other phycotoxins in Fishes and Shellfishes” should be corrected. I suggest the authors carefully check the similar errors.
5. The format of the references is not unified, such as ref.1 and ref.4. I suggest the authors uniform all the references.
Author Response
Reviewer #2 comments:
Due to the hazard of phycotoxins or marine toxins to humans, livestock, and pets, it is of significance to develop sensitive, specific and fast methods to detect them. Aptamer can identify specific target and is easy to synthesized, so it is commonly employed to establish assays for the detection of its target. This article reviews the designed aptasensors based on the different strategies for the detection of the various phycotoxins. The review is very comprehensive and will provide valuable support for the relevant research. Therefore, I recommend to publish this article after the following issues are addressed.
Q1. page 8, line 241, the authors cite ref.33 and show the method in Figure 1a. However, in the caption of Figure 1a,"Reprinted with permission from [55]." is used. This needs to be corrected, and I suggest the authors check all the similar situations in the whole manuscript.
Response: Thanks for your attention. We corrected it and checked all the similar situations in the whole manuscript. Please see Pages 10 and Line 331.
Q2. Page11 line 371 “However, any dual-mode aptasensor has not been fabricated for STX. So, developing dual-response aptasensors is novel for the sensitive detection of STX.” I wonder why there is no dual-mode aptasensor used for the detection of STX, is there any problem to solve?
Response: Authors are grateful for your attention. Actually, the authors intended to imply that no dual-mode aptasensor for this toxin (STX) has been developed based on experiments conducted to date. Consequently, the creation of dual response aptasensors may open up new possibilities for the accurate detection of STX. We revised it. Please see Pages 11 and Lines 357-368.
Q3. Figure 1 and Figure 4 are not very clear, and the words in these figures cannot be well discriminated.
Response: Thanks for your attention. We revised it. Please see Figure 1 and Figure 4.
Q4. Some typos, such as page7 line170 "dditionally"; page9 line284 " Figure 1c)"; page13 line487 “Fishe”; page16 line584 “Fishe”; page19 line685 “humans This”; page20 line741 “Aptasensing platforms on for the other phycotoxins in Fishes and Shellfishes” should be corrected. I suggest the authors carefully check the similar errors.
Response: Thanks for your attention and the authors are grateful for your kind point of view about this manuscript. We revised them. Please see page7 line168, page14 line 479, page16 line 551, page19 line 649 and page20 line 733. Likewise, the manuscript was checked in terms of typing problems.
Q5. The format of the references is not unified, such as ref.1 and ref.4. I suggest the authors uniform all the references.
Response: Thanks for your attention. The style of references is based on APA of Style, 16th edition which with Ende Note X8 software are set. According to this format, if the number of authors is up to 3, the names of all authors are mentioned, for more than 3 authors, et al. is used.

Reviewer 3 Report (New Reviewer)
The review titled "Recent Advances in Aptasensing Strategies for Monitoring Phycotoxins: Promising for Food Safety" attempts to describe the aptasensing strategies built to detect phycotoxins in food. The authors did a detailed analysis of the different methods and have provided the aptamer sequences as well, which would be very beneficial for potential readers. However there are a few things that needs attention before publishing:
1. Though the authors have made detailed analysis of the said methods and strategies, critical analysis needs to be done in view of why these methods have not been successful in the field or what drawbacks these said methods have. The in depth discussion at the end of the manuscript only gives a very general idea which does not delve deep into what drawbacks these individual methods have and what can be done to ameliorate them.
2. Some of the figures provided needs to be submitted in high resolution and aligned properly. For example, figure 3a and 4a are distorted to fit the final figure scale.
3. The language of the manuscript needs to be double checked for grammar as there are some instances where the meaning of the sentence could not be deciphered. The authors can take the help of some proofreading software to do so.
Author Response
Reviewer #3 comments:
The review titled "Recent Advances in Aptasensing Strategies for Monitoring Phycotoxins: Promising for Food Safety" attempts to describe the aptasensing strategies built to detect phycotoxins in food. The authors did a detailed analysis of the different methods and have provided the aptamer sequences as well, which would be very beneficial for potential readers. However there are a few things that needs attention before publishing:
Q1. Though the authors have made detailed analysis of the said methods and strategies, critical analysis needs to be done in view of why these methods have not been successful in the field or what drawbacks these said methods have. The in depth discussion at the end of the manuscript only gives a very general idea which does not delve deep into what drawbacks these individual methods have and what can be done to ameliorate them. Response: Authors are Response: Authors are grateful for your valuable opinion. We revised it.
In Table No. 1, Pages 4 to 6, the features, sensitivity, and some of the scores of each type of aptasensor are fully presented, including features of these individual methods and making their comparison easier. In any case, authors suggested using some versatile amplification methods, including hybridization chain reaction (HCR), rolling circle amplification (RCA), recombinase polymerase amplification (RPA), and polymerase chain reaction (PCR) that can improve the sensitivity of the aptasensors (page 22, lines 810-816), or the possibility of taking advantages of dual-response sensors with multicolor output signals can be efficient for multiple target detection (page 11, lines 357-368), combining colorimetric strategies with microfluidic arrays for achieving cost-effective, simple to detection targets, and naked-eye method (page 20, lines 697-703) and similar items on 527-535 lines.
Q2. Some of the figures provided needs to be submitted in high resolution and aligned properly. For example, figure 3a and 4a are distorted to fit the final figure scale.
Response: Thanks for your attention. We revised it. Please see Figure 3 and Figure 4.
Q3. The language of the manuscript needs to be double checked for grammar as there are some instances where the meaning of the sentence could not be deciphered. The authors can take the help of some proofreading software to do so.
Response: Thanks for your attention. We revised it. We tried to rewrite or improve the text in terms of grammar and fluency.

Reviewer 4 Report (New Reviewer)
Although the review is a significant addition to scientific literature, it needs careful editing for scientific concepts and English grammar.
The explanation provided for Figure 2C. is wrong. This is not a FRET effect, but a quenching effect. Quenching vs. FRET is a different concept. FRET is fluorescence resonance energy transfer from one fluorophore to other without the need for excitation of the latter due to the molecular proximity effect. The concept explained in figure 2C is simply a quenching effect.
Line#487: Spell check- Fish. Also, check for inconsistent capitalization throughout the manuscript. For example, “Monitoring of Phycotoxin pollutants in Fishe” in Line 487.
Line 770: The term aptasensor is misspelled
Authors should compare aptasensors for phycotoxins with other aptamers such as pesticides. I recommend a recent review article: https://doi.org/10.1016/j.teac.2022.e00184 and few studies by the group on malathion, fipronil, diazinon, and fenitrothion could be compared with phycotoxin-based sensing.
Line 838: The sentence ‘Detailed information of different some aptamers applied in the sensors’ is grammatically wrong. Delete ‘some’ from the sentence.
Overall, English needs attention.
Section 6. Delete ‘In-depth’ from ‘in-depth discussion and conclusion”.
Author Response
Reviewer #4 comments:
Although the review is a significant addition to scientific literature, it needs careful editing for scientific concepts and English grammar.
Q1. The explanation provided for Figure 2C. is wrong. This is not a FRET effect, but a quenching effect. Quenching vs. FRET is a different concept. FRET is fluorescence resonance energy transfer from one fluorophore to other without the need for excitation of the latter due to the molecular proximity effect. The concept explained in figure 2C is simply a quenching effect.
Response: Thanks for your attention. Ref N.45 (Cheng et al. 2018) is connected with (Figure 2c) which is totally different from ref N. 35 (Lee et al. 2019) (related to fluorescence resonance energy transfer (FRET). Both the article and picture number 2 (section c) were also highlighted in the texts. Please see Page 9, Lines 290-296, Page 12 and 13, Lines 416-422, and Figure 2 c.
- Lee, Eun-Hee, and Ahjeong Son. "Fluorescence Resonance Energy Transfer Based Quantum Dot-Aptasensor for the Selective Detection of Microcystin-Lr in Eutrophic Water." Chemical Engineering Journal 359 (2019/03/01/ 2019): 1493-501.https://doi.org/10.1016/j.cej.2018.11.027
- Cheng, Sheng, Bin Zheng, Dongbao Yao, Shenglong Kuai, Jingjing Tian, Haojun Liang, and Yunsheng Ding. "Study of the Binding Way between Saxitoxin and Its Aptamer and a Fluorescent Aptasensor for Detection of Saxitoxin." Spectrochimica Acta Part A: Molecular and Biomolecular Spectroscopy 204 (2018/11/05/ 2018): 180-87. https://doi.org/10.1016/j.saa.2018.06.036
Q2. Line#487: Spell check- Fish. Also, check for inconsistent capitalization throughout the manuscript. For example, “Monitoring of Phycotoxin pollutants in Fishe” in Line 487.
Response: Thanks for your attention. We revised it. Please see Pages 7, 14, and 16 in lines 168,479, and 551 respectively.
Q3. Line 770: The term aptasensor is misspelled
Response: Authors are grateful for your kind point of view about this manuscript and they are grateful for your helpful comments. We revised it. Please see Page 20, Line 733
Q4. Authors should compare aptasensors for phycotoxins with other aptamers such as pesticides. I recommend a recent review article: https://doi.org/10.1016/j.teac.2022.e00184 and few studies by the group on malathion, fipronil, diazinon, and fenitrothion could be compared with phycotoxin-based sensing.
Response: Authors are grateful for your valuable opinion and suggestion. According to the potential of the sources involved, the authors have attempted to categorize the aptasensors used to detect the most significant marine toxins in this review. This topic itself covered so many extremely broad areas that we frequently had to sift the resources. It is unquestionably a very excellent idea to compare several aptasensor kinds to find food poisons in a separate investigation.
Q5. Line 838: The sentence ‘Detailed information of different some aptamers applied in the sensors’ is grammatically wrong. Delete ‘some’ from the sentence.
Response: Authors are grateful for your valuable opinion. We revised it. Please see Page 22, Line 797.
Q6. Overall, English needs attention.
Response: Thanks for your attention. We revised it. We tried to rewrite and improve the text in terms of grammar and fluency.
Q7. Section 6. Delete ‘In-depth’ from ‘in-depth discussion and conclusion”.
Response: Authors are grateful for your valuable opinion. We revised it. Please see Page 22, Line 799.

Round 2
Reviewer 4 Report (New Reviewer)
No additional comments.
This manuscript is a resubmission of an earlier submission. The following is a list of the peer review reports and author responses from that submission.
Round 1
Reviewer 1 Report
This paper reviews aptamer sensors based on different strategies designed to detect various algal toxins. However, the overall description is too simplistic, with little independent thinking on its own level, and the paper should be rejected outright.
1. Line 81-93 (especially ending part of Introduction): How is this study different from others? How is this work distinguished from so many similar researches reported up to date! Add some descriptions at the end of Introduction.
2. Discussion is very much dispersive. It should be digested and integrated.
3. The pictures in each section of an overview article should be integrated and logical, but in your article the pictures are only one picture from the article you quote, which I think is too scattered and lacks a summary picture of the class of phycotoxins you are describing.
4. Mechanism of aptasensors must be discussed and compared with previously reported publications in detail.
Reviewer 2 Report
Overall it is a comprehensive review on aptamesors of marine toxins. Here, I have two suggestions on this manuscript before it is accepted.
1. It would be much reader friendly if the authors could put the table to the early section of this manuscript, instead of in the conclusion part.
2. The Figure 3 did not clearly represent why the electrochemical response changes after the target binding to the aptamer on the tetrahedron.
Reviewer 3 Report
line 37: domoic corrosive (DA), may be it should be domoic acid?
line 70-71: "these difficulties limit the widespread application of immunosensing platforms" but authors didn't say anything about immunosensing platforms, only about mouse using.
line 205: most likely means "recovery", not "recovery data"
table 1. Please add sensitivity of analysis ("analytical signal vs. concentration" slope)
Reviewer 4 Report
The authors have presented an interesting subject with an interest in the field of environmental monitoring with the application of aptasensors. Recently, biosensors have found many valuable applications for different analytical purposes. The aptamers exhibit distinct advantages such as simple and fast chemical synthesis, easy modification with different functional groups, good stability, and low cost. The authors have compiled recent reports, mainly focused on the detection of marine phycotoxins. However, there are many reviews in this field particularly, and the recent related reviews should be cited in the manuscript, so it seems that the idea is not new, and this article does not cover the topic impressively. Frequently, the next section does not follow the previous one, sometimes it is possible to get lost in the manuscript text. There is a need for scientific storytelling to attract the attention of potential readers but not in the case of this article. Nevertheless, the article has scientific soundness and introduces significant merit and is a good compilation of recent applications of aptasensor in phycotoxins sensing. It seems that the title does not correspond with the content and the idea of ​​the paper. The authors should make a scientific comment about the work rather than only listing references. I expected to know the latest strategies and trends in phycotoxin detection from an analytical point of view but in the article, the target molecules were listed along with information about the detection methods. The only attempt at a summary is found in the conclusions section and table 1. I appreciate the effort put into obtaining permission to reproduce the figures, but in my opinion, the manuscript needs additional work to prioritize the methods and emphasize the analytical aspect*. Some of my suggestions were listed below:
Lines 183-195: should be corrected to it is great, eliminate, high-cost, represents, a simple etc.
Lines 339-340: “carbon-based nanomaterials” sounds better
Table 1. needs unification, the first column should be larger
In my opinion, the manuscript is unsuitable for publication in its current form and requires major revision before any other consideration.
Similar articles:
1. Toxins 2018, 10, 427; doi:10.3390/toxins10110427
2. Mar. Drugs 2022, 20(3), 198; https://doi.org/10.3390/md20030198
*Majdinasab, M.; Marty, J.L. Recent Advances in Electrochemical Aptasensors for Detection of Biomarkers. Pharmaceuticals 2022, 15, 995. https://doi.org/10.3390/ ph15080995